# Prognostic Value of EMT Gene Signature in Malignant Mesothelioma

**DOI:** 10.3390/ijms24054264

**Published:** 2023-02-21

**Authors:** Licun Wu, Kosuke Yoshihara, Hana Yun, Saraf Karim, Nastaran Shokri, Fatemeh Zaeimi, H. S. Jeffrey Man, Amin Zia, Emanuela Felley-Bosco, Marc de Perrot

**Affiliations:** 1Latner Thoracic Surgery Research Laboratories, Division of Thoracic Surgery, Toronto General Hospital, Princess Margaret Cancer Research Centre, University Health Network, University of Toronto, Toronto, ON M5G 1L7, Canada; 2Institute for Research Promotion, Graduate School of Medical and Dental Sciences, Niigata University, Niigata 951-8510, Japan; 3Department of Bioinformatics and Computational Biology, University of Texas MD Anderson Cancer Center, Houston, TX 77030, USA; 4Department of Immunology, University of Toronto, Toronto, ON M5S 2E8, Canada; 5Dycode Bio, Co., Toronto, ON L6C 2R9, Canada; 6Laboratory of Molecular Oncology, Department of Thoracic Surgery, University Hospital Zurich, 8091 Zurich, Switzerland; 7Institute of Medical Science, University of Toronto, Toronto, ON M5G 2C4, Canada

**Keywords:** malignant mesothelioma (MESO), epithelial–mesenchymal transition (EMT), gene signature, multiomic analysis, prognosis

## Abstract

Malignant mesothelioma (MESO) consists of epithelioid, biphasic, and sarcomatoid subtypes with different epithelial–mesenchymal transition (EMT) phenotypes. We previously identified a panel of four MESO EMT genes correlating with an immunosuppressive tumor microenvironment and poor survival. In this study, we investigated the correlation between these MESO EMT genes, the immune profile, and the genomic and epigenomic alterations to identify potential therapeutic targets to prevent or reverse the EMT process. Using multiomic analysis, we observed that the MESO EMT genes were positively correlated with hypermethylation of epigenetic genes and loss of CDKN2A/B expression. MESO EMT genes such as *COL5A2*, *ITGAV*, *SERPINH1*, *CALD1*, *SPARC*, and *ACTA2* were associated with upregulation of TGF-β signaling, hedgehog signaling, and IL-2-STAT5 signaling and downregulation of the IFN-α and IFN-γ response. Immune checkpoints such as *CTLA4*, *CD274* (PD-L1), *PDCD1LG2* (PD-L2), *PDCD1* (PD-1), and *TIGIT* were upregulated, while *LAG3*, *LGALS9*, and *VTCN1* were downregulated with the expression of MESO EMT genes. *CD160*, *KIR2DL1*, and *KIR2DL3* were also broadly downregulated with the expression of MESO EMT genes. In conclusion, we observed that the expression of a panel of MESO EMT genes was associated with hypermethylation of epigenetic genes and loss of expression of *CDKN2A* and *CDKN2B*. Expression of MESO EMT genes was associated with downregulation of the type I and type II IFN response, loss of cytotoxicity and NK cell activity, and upregulation of specific immune checkpoints, as well as upregulation of the TGF-β1/TGFBR1 pathway.

## 1. Introduction

Mesothelioma (MESO) is an aggressive cancer associated with poor prognosis and limited therapeutic options. Besides asbestos, some nanomaterials such as carbon nanotubes have been shown to be potential carcinogens that could induce MESO in mice [1,2]. Despite a ban on asbestos in most industrialized countries, the incidence of MESO continues to rise due to the long latency period after exposure [3]. Nowadays, encouraging progress has been made in the treatment of this disease [4]. Preclinical and clinical studies demonstrate that novel approaches using immunotherapy can enhance the impact of MESO treatment and improve survival [5,6,7]. Triple-modality therapy combining non-ablative radiation with dual immunotherapy and surgery can maximize the antitumor immune responses in a mouse model of mesothelioma [6]. A multicenter phase 3 trial demonstrated that first-line nivolumab plus ipilimumab improved survival compared to chemotherapy alone in unresectable pleural MESO [7]. However, the overall efficacy of these current treatment strategies remains limited so that the prognosis is still fairly poor [8].

MESO is characterized by a unique morphology, including epithelioid, sarcomatoid, and biphasic subtypes, providing an optimal model to study epithelial–mesenchymal transition (EMT) phenotypes [9]. Considerable evidence has shown that EMT contributes to tumor immune escape, angiogenesis, invasion and metastasis, stemness, and therapy resistance, thus leading to critical impacts on patient survival [10,11].

Rapid development of multiomic approaches and bioinformatics made it possible to investigate particular genes or gene signatures systemically from genome to transcriptome, proteome, metabolome, and epigenome. There has been a huge body of publications in omics in the most recent two decades [12]. Multiomics data analysis is an important tool for cancer molecular biology studies that has led to breaking discoveries. The interactions between transcriptome and proteome complexity have been investigated serially by temporal dynamics and spatiotemporal dynamics [13,14].

Using the EMTome platform, we identified a panel of MESO EMT genes in mouse models that were confirmed to be present and of importance in human MESO cell lines and patient samples [15,16]. This MESO EMT gene panel is a poor prognostic indicator in MESO patients [16]. Defining the association of this gene panel with the immune profile, as well as the genomic and epigenomic alterations, could provide important information on how to prevent or reverse the EMT process in MESO.

In this study, we therefore investigated the correlation between the different EMT genes from the panel and the immune cells in the tumor microenvironment, the expression of specific immune check points, the stromal score, as well as the genomic and epigenetic alterations in MESO.

## 2. Results

### 2.1. EMT Gene Signature Associated with Overall Survival in Mesothelioma

We identified nine EMT genes, *COL5A2, ITGAV*, *SERPINH1, CALD1, TIMP3, SPARC, ACTA2, TNFRSF12A,* and *TPM2,* associated with survival in the MESO TCGA (Figure 1B). Survival curves demonstrated that *SERPINH1*, *SPARC*, *COL5A2*, *TNFRSF12A*, and *ACTA2* were the most significant genes, leading to an early separation of the survival curves, while CALD1 and TIMP3 had delayed separation of the survival curves after 30 months only (Figure 1C). We had previously identified that four of these genes (*COL5A2*, *SPARC*, *ITGAV*, and *ACTA2*) were strongly associated with EMT^16^. However, in this study, we observed that the prognostic value increased with a greater number of genes analyzed (Appendix A).

### 2.2. Gene Expression of Nine EMT Gene Signatures in Tumors, Pleural Effusion, and TCGA Database

In the TCGA data set, the expression of the nine EMT genes was significantly higher in non-epithelioid MESO than in epithelioid MESO (Figure 2A). scRNA-Seq acquired from untreated MESO patients with pleural effusion (Figure 2B,C) and biopsy tumor tissues (Figure 2D,E) demonstrated that these nine EMT genes were overexpressed predominantly in mesothelioma cells characterized by *MSLN* expression. A second cluster expressing some of these genes was detected in the tumor tissues (Figure 2E). This second cluster could represent a cancer-associated fibroblast (CAF) population characterized by low expression of *MSLN* and *WT1* and high expression of *ACTA2*, *S100A4*, *COL5A2*, *SPARC,* and *FN1* (Appendix A).

### 2.3. EMT Score and Stromal Score Are Associated with Survival in MESO TCGA Database

The MESO EMT score calculated from our EMT gene signature significantly correlated with overall survival in the TCGA database. For comparison, the EMT score calculated from PanCancer EMT genes did not reach statistical significance, suggesting that the MESO EMT score may be more relevant for MESO (Figure 3A–C). Stromal score is also known to display significant prognostic value [17]. As the stromal components play a critical role in the EMT process and immunosuppressive microenvironment, we compared the epithelioid and non-epithelioid subtypes, and found significantly higher values of stromal scores in non-epithelioid MESO (Appendix A). High stromal score was associated with worse survival (Figure 3D). Interestingly, the MESO EMT scores positively correlated with the stromal score (Figure 3E). Since cancer-associate fibroblasts (CAF) are a major cell type contributing to the EMT process, we analyzed whether the MESO EMT gene expression correlated with CAF enrichment in MESO using Timer2.0 and observed a strong correlation in different analytic methods (Appendix A).

Altogether, this analysis demonstrated that the MESO-EMT score was strongly associated with overall survival in MESO patients and positively correlated with stromal score. CAF enrichment correlated with the expression of the EMT genes, supporting the importance of this cell population in the EMT process.

### 2.4. Proteomic Analysis and Signaling Pathway Networks of MESO EMT Genes

Generic protein-to-protein interaction (PPI) mapped the MESO EMT genes to the corresponding molecular interaction database. The process typically displays one big subnetwork with several smaller islands. Network view of the interactome was shown as nodes and PPIs as edges connecting the nodes (Appendix A). GO enrichment analysis for the MESO EMT genes in biological process (BP), cellular component (CC), and molecular function (MF) terms, and top ten enriched GO terms demonstrated the role of these genes in the cytoskeleton and cell matrix (Figure 4A).

Hallmark gene sets demonstrated that the top upregulated pathways correlating with expression of the MESO EMT genes were TGF-β signaling, tumor angiogenesis, hedgehog signaling, IL-2-STAT5 signaling, KRAS, and hypoxia pathways (Figure 4B). Further analysis confirmed the strong positive correlation between the EMT genes and the expression of TGFB1 and TGFBR1 (Appendix A). Expression of *COL5A2*, *ITGAV*, *SERPINH1*, *CALD1*, *SPARC*, and *ACTA2* were associated with downregulation of the interferon-α and interferon-γ response (Figure 4B), as well as downregulation of activated CD8 tumor infiltrating lymphocytes (Act CD8), which induced dendritic cells (iDC) and immature B cells (imm B) in the tumor microenvironment (Figure 4C). Expression of MESO EMT genes was associated with upregulation of immune suppressive markers such as *CTLA4*, *CD274* (PD-L1), *PDCD1LG2* (PD-L2), *PDCD1* (PD-1), and *TIGIT* and downregulation of *LAG3*, *LGALS9*, and *VTCN1* (Figure 4C,D). *CD160*, *KIR2DL1*, and *KIR2DL3* were also broadly downregulated with the expression of MESO EMT genes, emphasizing the potential impact of the EMT process on NK cell activity (Figure 4D).

### 2.5. Correlation of MESO EMT Genes with Genomic/Epigenomic Alterations

Genomic analysis of the MESO EMT genes showed that mutations were extremely rare in these genes, supporting their importance (Appendix A). Expression of *CDKN2A* and *CDKN2B* negatively correlated with the expression all nine EMT genes, while the epigenetic gene expression positively correlated with the MESO EMT gene expression (Figure 5). The altered genes were hypomethylated, with the exception of *PTCH1*, whereas the epigenetic genes were mostly hypermethylated with the exception of *DNMT3A*, *HDAC2,* and *HAT1*. Most of the MESO EMT genes were also hypermethylated, with the exception of *ITGAV* and *TNFRSF12A* (Appendix A).

All together, these data suggest that the EMT process is associated with hypermethylation of key epigenetic genes and the loss of expression of *CDKN2A* and *CDKN2B*.

### 2.6. Low CDKN2A/B Expression and Homozygous Deletion (HD) of p16 Had an Inverse Outcome on the Patients with MESO

The lack of expression of *CDKN2A* and *CDKN2B* in MESO is driven by homozygous deletion (HD) and heterozygous deletion in MESO (Figure 6A). Low expression of *CDKN2A* and *CDKN2B* are associated with worse survival in the MESO TCGA data (Figure 6B,C). Similarly, p16^INK4a^ HD (*CDKN2A*) positivity and p15^INK4b^ HD (*CDKN2B*) positivity are associated with worse survival in MESO (Figure 6D,E). The expression of EMT genes is associated with lack of expression of *CDNK2A* and *CDNK2B* and downregulation of interferon genes (Appendix A).

## 3. Discussion

Considerable evidence has shown that EMT plays critical roles in tumor angiogenesis, tumor progression, and metastasis, leading to poor prognosis. Therefore, EMT genes have been used to predict survival using the PanCancer EMT score across different types of tumors. The PanCancer EMT signature, however, was generated from patient-derived xenograft and identified global molecular alterations, but these alterations were not necessarily specific for MESO [18]. Our study specifically focused on MESO to identify a panel of EMT genes correlating with survival.

MESO provides an EMT spectrum ranging from a predominantly epithelial phenotype in the epithelioid MESO to a predominantly mesenchymal phenotype in the sarcomatoid MESO [19]. Some EMT transcription factors (e.g., Slug, Twist, ZEB1, and ZEB2) are known to be upregulated in sarcomatoid MESO [20]. However, these transcription factors are difficult to use routinely as prognostic markers and are not easily targetable to reverse the EMT process. We therefore focused on identifying specific EMT genes in MESO that were potentially readily available as biomarkers and could be used as therapeutic targets.

In our previous work [16], we screened the list of 23 genes that were involved in the Hallmark EMT pathway using the gene set enrichment analysis (GSEA) MSigDB database. The top four genes (*COL5A2*, *ITGAV*, *SPARC*, and *ACTA2*) associated with the strongest correlation with EMT Top50 mesenchymal genes were selected for further analysis, confirming their prognostic importance in mesothelioma and other solid tumors using the TCGA database of online platforms.

In the current study, we refined the analysis by narrowing down this gene list using the EMTome platform. Sixteen genes were positively correlated with the mesenchymal phenotype, while negatively correlated with the epithelial phenotype in the EMTome, and finally, nine genes were found to be associated with survival differences using GEPIA in the MESO TCGA cohort, as indicated in the diagram (Figure 1A). This nine gene signature had significant prognostic value in MESO.

*SERPINH1*, *SPARC*, *TNFRSF12A*, *COL5A2*, and *ACTA2* were associated with an early separation of the survival curves and could potentially be relevant EMT biomarkers and targets. These EMT genes were also associated with TGFB1/TGFRB1 signaling, hedgehog signaling, and IL-2-STAT5 signaling, emphasizing the importance of these pathways in the EMT process. These results corroborate previous studies showing that the mesenchymal phenotype may be a therapeutic target in MESO by suppressing lysine-specific demethylase 1 (LSD1) [21]. The TGF-β pathway plays an important role in the EMT process and could thus provide potential therapeutic targets [22,23,24].

The role of TGF-β as a primary inducer of EMT has been reported in several cancers, while other studies reported a more ambivalent role for TGF-β1 [25]. In mesothelioma, Creaney et al. recently demonstrated through comprehensive genomic analysis and tumor immune profiling that overexpression of TGFB1 contributed to the immunosuppressive microenvironment and was associated with worse survival in MESO [26]. TGFB1 can also enhance PD-L1 expression and treatment with a TGF-β1 receptor kinase inhibitor can downregulate PD-L1 expression in head and neck cancer, suggesting that targeting the EMT process could reverse the immunosuppressive tumor microenvironment [27]. However, more mechanistic studies will be important, as a more ambivalent role of TGFB1 has been seen in some other studies [28,29,30]. In our study, we observed that overexpression of the EMT genes was associated with downregulation of type I and type II IFN, downregulation of activated CD8 TILs, and induced DC in the tumor microenvironment. We also observed potent downregulation of CD160, a marker of cytolytic effector activity expressed by NK cells and CD8 T cells. The EMT gene expressions were associated with the upregulation of specific immune checkpoints, such as *CTLA4*, *CD274*, *PDCD1LG2*, *PDCD1*, and *TIGIT*, and downregulation of *LAG3*, *LGALS9*, and *VTCN1*. This finding corroborates a previous observation demonstrating the presence of immune exhaustion markers associated with the EMT process [23].

Evidence suggests that the interaction between cancer-associated fibroblasts (CAF) and cancer cells is important to support the EMT process and cancer cell stemness [31]. CAFs are a critical player in maintaining the stromal matrix and could be a novel therapeutic target [32,33]. In the scRNA-Seq analysis, we found that the CAF cluster expressed a high level of some EMT genes, such as *COL5A2*, *FN1*, and *SPARC*, suggesting that these cells directly contribute to the EMT process. CAFs isolated from lung cancer promoted EMT via stromal cell-derived factor-1 (SDF-1), thereby upregulating CXCR4 and β-catenin [34]. Therefore, targeting the CAF-derived SDF-1-mediated CXCR4/β-catenin cascade may be an effective approach for lung cancer treatment [34]. Another approach to target EMT is metabolic reprogramming, which seems very encouraging [35].

To better understand the genomic and epigenomic alterations driving MESO development as potential targets to modulate the EMT process, we investigated the molecular correlations between EMT gene expression and genomic, as well as epigenomic, alterations in MESO [36]. Comprehensive genomic analysis of MESO demonstrated recurrent mutations, gene fusions, and splicing alterations in MESO across different trials [37,38,39]. Gene mutations and deep deletions are likely to be the underlying mechanisms resulting in gene inactivation of *NF2, BAP1,* and *SETD2*. Additionally, genomic alterations in Hippo, mTOR, histone methylation, and p53 signaling pathways may also interact with epigenomic alterations. Interestingly, EMT genes themselves had low rates of mutation, but their expression was strongly correlated with genomic alterations, especially C*DKN2A* and *CDKN2B,* which are frequently altered with predominantly homozygous deletion in MESO.

Recent insights in epigenetic alterations provide novel therapeutic options in MESO [40]. Epigenetic modifications are of potential importance as they can modulate the EMT pathway [41]. Considerable research has been devoted to understanding the epigenetic mechanisms on promoter methylation and gene silencing during MESO carcinogenesis after asbestos exposure. The importance of epigenetics in the EMT process is supported by our findings, demonstrating the high rate of epigenetic hypermethylation in association with EMT genes’ upregulation. Taken together, a better understanding of EMT gene profiling provided by this study will help to establish the linkage between EMT and epigenetic mechanisms. Hence, a promising field of drug discovery to modulate the EMT process is focusing on epigenetic targets. This approach may also optimize the effects of immunotherapy [42,43,44].

In conclusion, EMT gene overexpression is an unfavorable prognostic factor for MESO. Multiomic analysis indicated that EMT gene overexpression correlated with the loss of CDKN2A and CDKN2B expression and epigenomic alterations. The immune characteristics associated with the EMT gene expression suggest that the EMT process orchestrates the immunosuppressive tumor microenvironment. EMT genes contribute to a wide variety of hallmark pathways, including TGFB1 signaling, hedgehog signaling, and IL-2-STAT5 signaling, which could be targeted to prevent or reverse the process. Altogether, overexpression of EMT genes could drive mesothelioma cells to achieve mesenchymal phenotypes during tumor progression, thus promoting tumor angiogenesis, invasion, and metastasis, stemness, and therapy resistance; therefore, this EMT gene signature could have a potential role as a prognostic biomarker and provide potential therapeutic targets in MESO.

## 4. Materials and Methods

### 4.1. Murine Mesothelioma Cells and Mouse Model

Murine mesothelioma cell line RN5 was derived from C57BL/6 mice after exposure to asbestos [45]. RN5 cells are characterized by biphasic morphology [46]. The methods of murine models were described previously [16].

### 4.2. Patients with Mesothelioma

Pleural effusions (PE, *n* = 3, all with epithelioid subtypes) and biopsy tumor tissues (*n* = 5, 1 biphasic and 4 epithelioid subtype) were acquired at the time of diagnosis. Samples were collected from different patients, except for one patient providing both PE and tumor biopsy. All samples were collected between 2019 and 2020. This study was approved by our institution, University Health Network (REB#19-5858), and all patients signed the REB-approved consent form. Single cells from pleural effusion and tumor tissues were run for scRNA-Seq [6,16]. Briefly, cell pellets obtained from PE were passed through a cell strainer (Φ70µm) and washed thrice with PBS. Tumor tissues were cut into approximately 5 mm cubes and washed with PBS, then transferred to a gentle MACS^TM^ C tube (Miltenyi Biotec, Bergisch Gladbach, Germany) by adding 4.5 ml HBSS, 80 µL DNase I, and 400 µL Collegenase A. Gentle MACS Dissociator was used to run an 11 min protocol. Single cells were filtered with a cell strainer and washed thrice with PBS. Single cells from patients were also processed by Princess Margaret Genomic Centre, UHN, Toronto, ON, Canada.

### 4.3. Identification of MESO EMT Gene Signature

The EMT gene panel was generated by: (1) selecting the genes that were upregulated at each time point (week 1, 2, 3, 4, 5, 8) on microarray and at week 4 on scRNA-Seq after tumor cell injection in mice; (2) then overlapping these genes with the Hallmark EMT gene set using the gene set enrichment analysis (GSEA) MSigDB database [16] and GeneVenn, which is available for free at https://www.bioinformatics.org/gvenn/ (accessed on 1 May 2021); (3) then selecting genes that were positively correlated with the mesenchymal phenotype and negatively correlating with the epithelial phenotype in the EMTome platform; and (4) finally selecting the genes that were impacting survival using the gene expression profiling interactive analysis (GEPIA) in the TCGA MESO (Figure 1A).

### 4.4. Online Resources for Multiomics Analysis

Survival analysis was carried out utilizing the analytical tool GEPIA2 (http://gepia2.cancer-pku.cn/ accessed on 1 May 2021). GEPIA is a web server for cancer and normal gene expression profiling and interactive analysis. GEPIA2 is able to deliver fast and customizable functionalities based on TCGA (The Cancer Genome Atlas) and genotype-tissue expression data; Mesothelioma TCGA data were retrieved from the cBioPortal for cancer genomics (https://www.cbioportal.org/datasets accessed on 1 January 2020). The prognostic value of the expression of EMT genes (*COL5A2, ITGAV*, *SERPINH1, CALD1, TIMP3, SPARC, ACTA2, TNFRSF12A,* and *TPM2*) was evaluated using the online databases^16^. To analyze the overall survival (OS), we set the median expression as the threshold by which patient samples were divided into high and low expression groups and used a Kaplan–Meier survival plot with the hazard ratio (HR), a 95% confidence interval (CI), and a log-rank test *p* value.

Loupe Cell browser (https://support.10xgenomics.com/single-cell-gene-expression accessed on 1 January 2020, https://crescent.cloud/ accessed on 1 January 2020) was employed to identify the EMT gene expression in tumors and PE from MESO patients. EMTome was used to identify the EMT gene signature associated with mesenchymal phenotype in MESO (http://www.emtome.org/ accessed on 1 January 2020). Gene set enrichment analysis, GSEA (https://www.gsea-msigdb.org/gsea/msigdb/ accessed on 1 May 2021), was used to identify the hallmark EMT gene sets. TIMER2.0 (http://timer.cistrome.org/ accessed on 1 May 2021) was exploited for systematic analysis of immune infiltrates across diverse cancer types. In addition, we used ESTIMATE (https://bioinformatics.mdanderson.org/estimate/ accessed on 1 May 2021) to score for tumor purity, the level of stromal cells present, and the infiltration level of immune cells in tumor tissues based on expression data. Tumor-immune system interactions, TISIDB (http://cis.hku.hk/TISIDB/ accessed on 1 May 2021), was also employed. Comprehensive gene expression profiling and network visual analytics was achieved with NetworkAnalyst (https://www.networkanalyst.ca/ accessed on 1 January 2020).

### 4.5. Calculation of EMT Score and Stromal Score

MESO EMT score was calculated based on the average z score of mesenchymal gene expression subtracted by the average z score of epithelial gene expression [47]. The EMT score was generated from the TCGA data set.

PanCancer EMT score was calculated by the average z score of pan-cancer mesenchymal gene expression subtracted by the average z score of epithelial gene expression.

Stromal scores were calculated by Dr. Kosuke Yoshihara, based on the protocol using the platform (https://bioinformatics.mdanderson.org/estimate/ accessed on 1 May 2021) that was developed by MD Anderson Cancer Center.

### 4.6. Selection of Genes of Interest with Genomic/Epigenomic Alterations

Genomic and epigenomic alterations are important factors in MESO [48]. We selected specific genes that are known to be altered in MESO to analyze their correlation with the MESO EMT genes. Selected genes included *BAP1, NF2, TP53, LATS2, SETD2, CDKN2A, CDKN2B, PTCH1, KRAS,* and *EGFR* [49,50,51]. The gene list of epigenomic alterations included *EZH2, DNMT1, TRDMT1 (DNMT2), DNMT3A, DNMT3B, WHSC1 (NSD2), TET1, HDAC2,* and *HAT1* [52].

### 4.7. Statistical Analysis

Statistical analysis was performed with GraphPad Prism 8.0 (GraphPad Inc. San Diego, CA, USA). Unpaired two-tailed Student’s *t* test was used to compare two groups. OS was compared using the log-rank test. All tests were two-tailed and a *p* value < 0.05 was considered significant. Results were presented as mean ± SEM. *, *p* < 0.05; **, *p* < 0.01; ***, *p* < 0.001 in all figures.

## Figures and Tables

**Figure 1 ijms-24-04264-f001:**
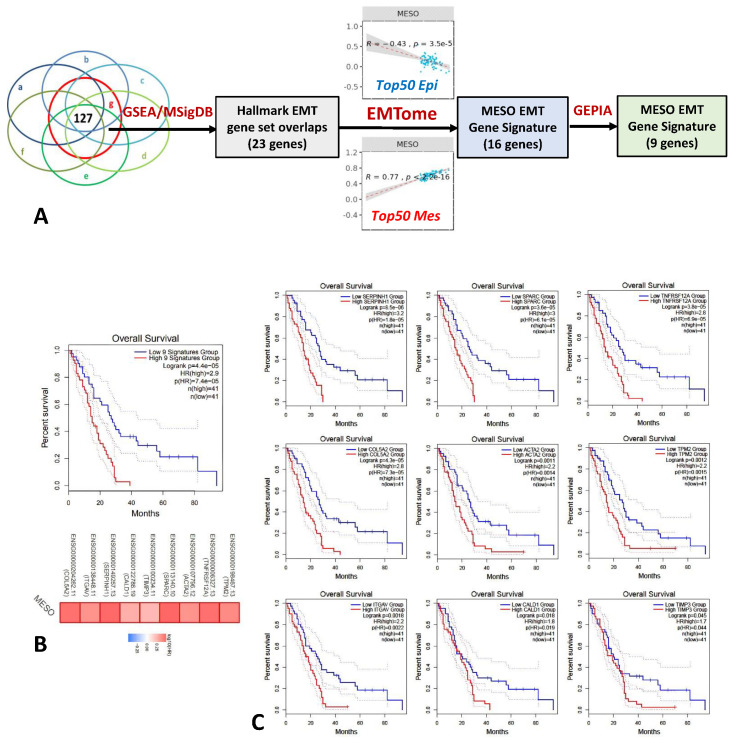
Mesothelioma EMT gene signature. (**A**) Twenty-three genes were identified by overlapping upregulated genes after tumor cell injection in mice and the Hallmark EMT gene set using the gene set enrichment analysis (GSEA) MSigDB database^16^, 16 genes then positively correlated with the mesenchymal phenotype and negatively correlated with the epithelial phenotype in the EMTome, and finally, 9 genes were associated with survival differences using GEPIA in the MESO TCGA cohort. (**B**) Survival curve based on expression level of these 9 EMT genes demonstrated significant survival differences in MESO TCGA (Logrank *p* = 4.4 × 10^−5^; Note: 4.4×10^-5^ is switched automatically into 4.4e-5 by the program.). (**C**) Survival curves based on the expression level of each EMT gene starting from the most significant gene, *SERPINH1* (Logrank *p* = 8.5 × 10^−6^), to the least significant gene, *TMP3* (Logrank *p* = 0.045), in MESO TCGA. Note that *COL5A2*, *ITGAV*, *SPARC*, and *ACTA2* had previously been found to be important EMT genes associated with survival in the TCGA database [16].

**Figure 2 ijms-24-04264-f002:**
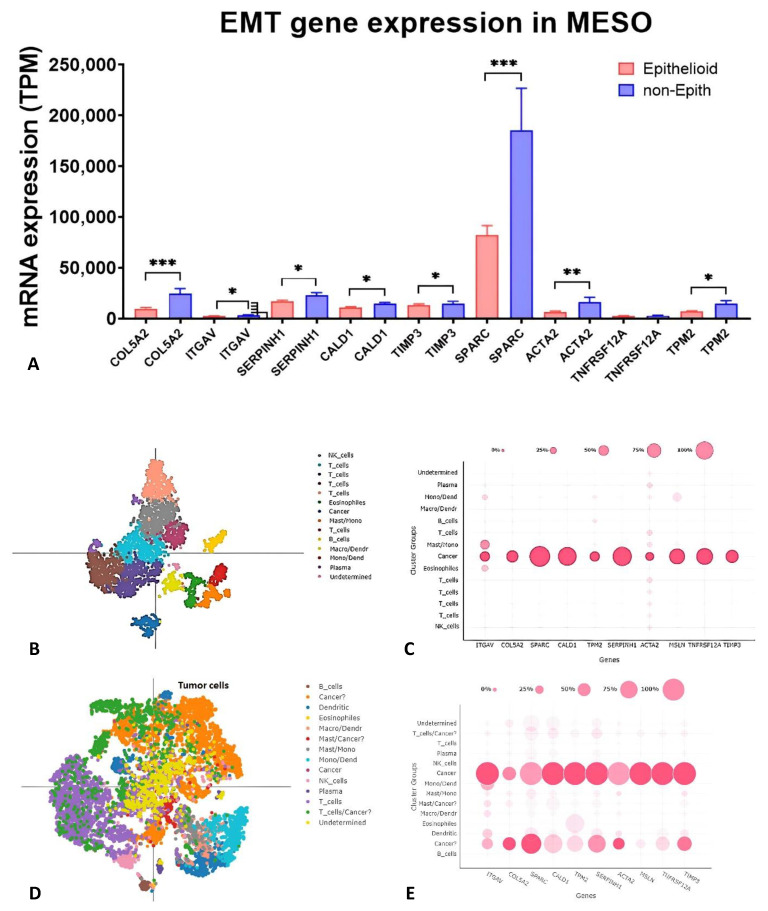
Gene expression of 9 EMT gene signature. (**A**) In the TCGA data set, the EMT gene expression was significantly higher in non-epithelioid MESO than epithelioid MESO. (**B**,**C**) scRNA-Seq results acquired from MESO patients with pleural effusion (*n* = 3) and (**D**,**E**) biopsy tumor tissues (*n* = 5). EMT genes were expressed predominantly on mesothelioma cells which were characterized by MSLN expression in both pleural effusion and biopsy tumor tissues. * *p* < 0.05, ** *p* < 0.01, *** *p* < 0.001.

**Figure 3 ijms-24-04264-f003:**
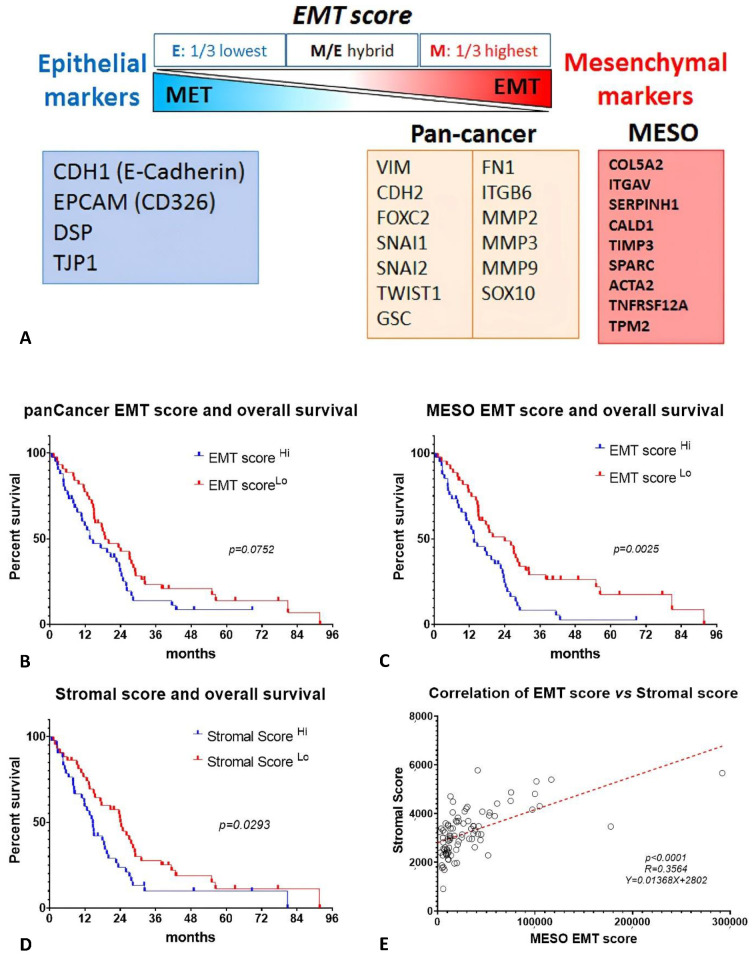
MESO EMT scores and stromal score are associated with survival in MESO TCGA database. (**A**) Diagram illustrating the EMT process and markers used to calculate the PanCancer EMT score and MESO EMT score; (**B**) overall survival according to the PanCancer EMT score (EMT score ^Hi^ *n* = 43, EMT score ^Lo^ *n* = 44); (**C**) overall survival according to the MESO EMT score (MESO EMT score ^Hi^ *n* = 43, MESO EMT score ^Lo^ *n* = 44); (**D**) overall survival according to the stromal score (Stromal score ^Hi^ *n* = 43, Stromal score ^Lo^ *n* = 44), and (**E**) correlation of MESO EMT score with stromal score.

**Figure 4 ijms-24-04264-f004:**
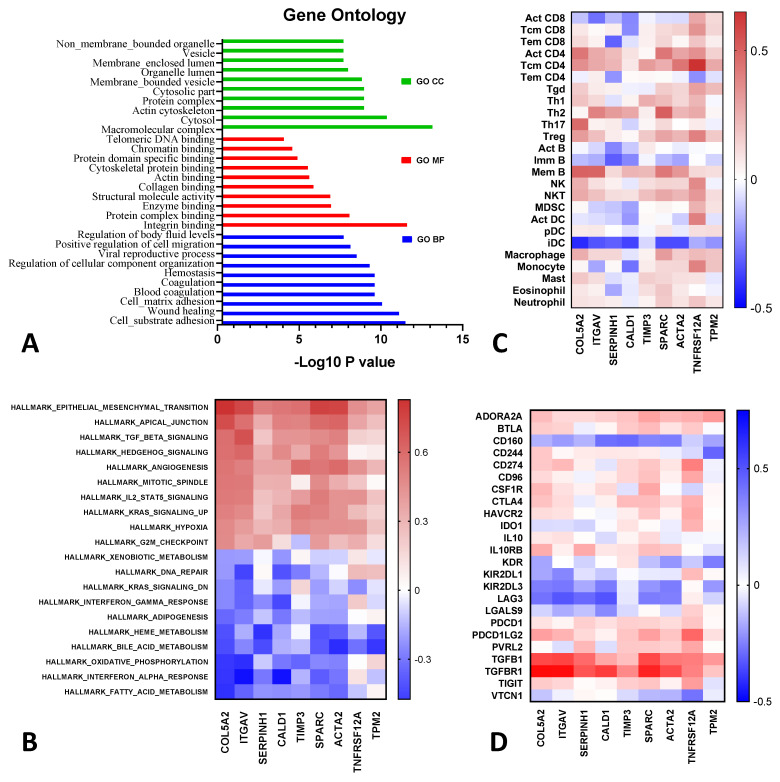
Gene ontology and heatmap correlating the MESO EMT genes with the immune profile. (**A**) Gene ontology (GO) enrichment analysis for the MESO EMT genes in biological process (BP), cellular component (CC), and molecular function (MF). (**B**) Hallmark enrichment gene sets positively and negatively correlating with the MESO EMT genes by NetworkAnalyst. (**C**) EMT gene expression correlating with tumor-infiltrating lymphocytes (TIL) in heatmap. (**D**) Correlation between EMT gene expression and immune exhaustion markers in heatmap.

**Figure 5 ijms-24-04264-f005:**
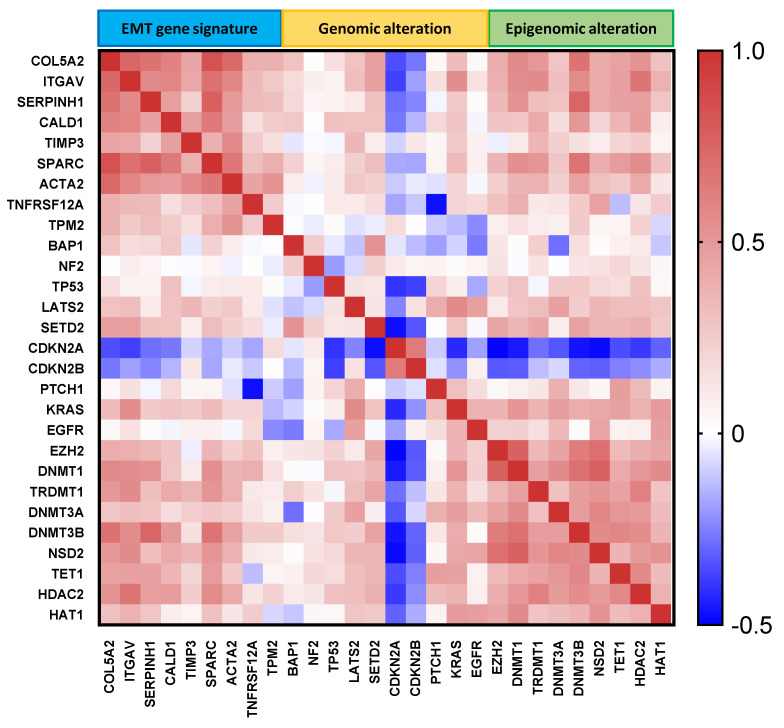
MESO EMT gene expression correlated with genomic/epigenomic alterations. Red and blue colors represent positive and negative correlation, respectively. The bar on the right represents the *p* values of correlation.

**Figure 6 ijms-24-04264-f006:**
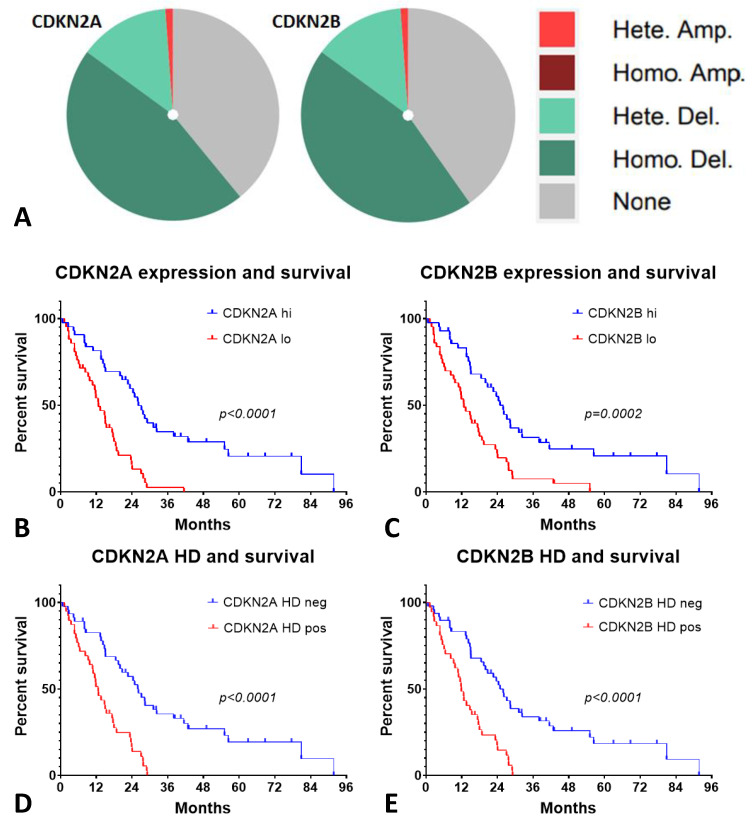
*CDKN2A/B* genomic alterations in TCGA database *CDKN2A* expression and homozygous deletion had an inverse effect on survival in MESO. (**A**) Homozygous deletion (HD) and heterozygous deletion of *CDKN2A* and *CDKN2B* genes are one the most frequent alterations in MESO; (**B**,**C**) lower expression of *CDKN2A* and *CDKN2B* genes are associated with worse survival in MESO; (**D**,**E**) homozygous deletion (HD) positivity of *CDKN2A* or *CDKN2B* are associated with worse survival in MESO.

## Data Availability

All data were included in this manuscript. No new data were created.

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
