# Peer review of "Prognostic Value of EMT Gene Signature in Malignant Mesothelioma"

_ijms, 2023, doi:10.3390/ijms24054264_

Round 1
Reviewer 1 Report (Previous Reviewer 1)
The Authors have successfully re-revised their MS, which is suitable for publication.
Author Response
Author's Reply to the Review Report (Reviewer 1)
Comments and Suggestions for Authors:
The Authors have successfully re-revised their MS, which is suitable for publication.
-Thank you very much for your encouraging comment.
Reviewer 2 Report (New Reviewer)
The study identified EMT gene signature and their prognostic value in mesothelioma. The gene signature was identified from mesothelioma mouse model and validated in the TCGA data set.
In general, it is a very interesting concept and EMT markers may be useful for identifying new treatment target.
While the correlation between the genes and stromal score, genome and epigenome are novel, some data such as prognostic values of some genes in figure 1 (survival curves from COL5A2, SPARC etc.) and the association with immune cells in figure 4 (for COL5A2 but in a different format) were already analyzed and presented. The author should thus emphasize what has been already performed and what is novel in the current manuscript for the clarity of the manuscript.
In the previous publication, they identified and selected 4 EMT associated genes COL5A2, ITGAV, SPARC and ACTA2. In the current manuscript, they aimed to find association between genes from the panel and the immune cells in the tumor microenvironment, the expression of specific immune check points, the stromal score as well as the genomic and epigenetic alterations in MESO. Nevertheless, in the current manuscript, additional 5 genes have been added. For the clarity, they should also explain what has been done differently for data analysis so that now a total of 9 genes are included.
The method with mouse model (2.1) looks identical to their previous publication and there is no data from this part except the gene signature. I think it is more appropriate to cite the original paper and not repeating the method as if it was performed here.
Author Response
Author's Reply to the Review Report (Reviewer 2)
Comments and Suggestions for Authors:
The study identified EMT gene signature and their prognostic value in mesothelioma. The gene signature was identified from mesothelioma mouse model and validated in the TCGA data set.
In general, it is a very interesting concept and EMT markers may be useful for identifying new treatment target.
While the correlation between the genes and stromal score, genome and epigenome are novel, some data such as prognostic values of some genes in figure 1 (survival curves from COL5A2, SPARC etc.) and the association with immune cells in figure 4 (for COL5A2 but in a different format) were already analyzed and presented. The author should thus emphasize what has been already performed and what is novel in the current manuscript for the clarity of the manuscript.
In the previous publication, they identified and selected 4 EMT associated genes COL5A2, ITGAV, SPARC and ACTA2. In the current manuscript, they aimed to find association between genes from the panel and the immune cells in the tumor microenvironment, the expression of specific immune check points, the stromal score as well as the genomic and epigenetic alterations in MESO. Nevertheless, in the current manuscript, additional 5 genes have been added. For the clarity, they should also explain what has been done differently for data analysis so that now a total of 9 genes are included.
- In our previous work16, we screened the list of 23 genes that were involved in the Hallmark EMT pathway using the gene set enrichment analysis (GSEA) MSigDB database. Four top genes (COL5A2, ITGAV, SPARC and ACTA2) associated with the strongest correlation with EMT Top50 mesenchymal genes were selected for further analysis confirming their prognostic importance in mesothelioma and other solid tumors using the TCGA database of online platforms.
In the current study, we refined the analysis by narrowing down this gene list using the EMTome platform. Sixteen genes were positively correlated with the mesenchymal phenotype, while negatively correlated with the epithelial phenotype in the EMTome, and finally 9 genes were found to be associated with survival differences using GEPIA in the MESO TCGA cohort, as indicated in the diagram (Fig 1A). This 9-gene signature had significant prognostic value in MESO.
This point was mentioned in the results section (line 172-174). Each gene has potential to be a valuable marker for prognosis and a therapeutic target, which is clinically relevant. We added this statement to the discussion section on page 13 lines 332-349.
The method with mouse model (2.1) looks identical to their previous publication and there is no data from this part except the gene signature. I think it is more appropriate to cite the original paper and not repeating the method as if it was performed here.
-We agree with you that the method has some identical descriptions with the previous publication. The initial idea was to include this method to facilitate the reading by describing how this gene signature was generated. This part has been removed and replaced by a concise statement on page 2 lines 78-92: The method of murine models was described
Round 2
Reviewer 2 Report (New Reviewer)
The authors have sufficiently answered the questions regarding comparisons with the previous publication and provided the explanation also in the manuscript.
With these clarifications, I believe the manuscript has sufficient quality for publication in IJMS in the present form.
This manuscript is a resubmission of an earlier submission. The following is a list of the peer review reports and author responses from that submission.
Round 1
Reviewer 1 Report
This is an interesting MS showing strong correlation between the expression of a 9 EMT gene signature and poor prognosis in malignant mesothelioma (abbreviated as MESO by the Authors). The study in large part is a follow-up of a recent publication by the same research group (reference xvi). The MESO EMT gene signature was defined by transcriptomics analysis of overlapping up-regulated genes (as assessed by GeneVenn diagrams) in a mouse model (evaluated by microarrays) and in pleural effusions of MESO patients from the authors’ institution (evaluated by scRNA-seq). Utilizing multiomics analyses and several databases, the Authors display the prognostic value and correlation with OS of the MESO EMT gene signature, which cells express the 9 EMT genes (allegedly MESO cells and CAFs), and that EMT gene expression is higher in non-epithelioid than epithelioid MESO samples of the TCGA db. Moreover, to explain the negative prognostic impact of the 9 EMT gene signature, they show that it correlates with the hypermethylation of certain epigenetic genes and with the loss of CDKN2A/B expression, and that it is linked to important cancer-relevant pathways and an immunosuppressive microenvironment.
However, some of the methods and data need clarifications and adjustments before the MS is fully suitable for publication.
SPECIFIC POINTS
MATERIALS AND METHODS
As remarked by the Authors, MESO is a cancer type with strong tendency to EMT, as proven by its 3 main histological subtypes, including the sarcomatoid with fully occurred EMT. Thus, the study carried out by Wu et al. is certainly relevant both for the understanding of EMT pathogenesis in cancer in general and in MESO, and for identifying possible vulnerabilities in this process that may represent novel potential therapeutic targets. In this respect, a more detailed histopathological description of the utilized samples is missing. It is only mentioned that the murine MESO cell line RN5 had biphasic morphology. Did the tumors that grew in mice after i.p. injection maintain a biphasic histology or did they become sarcomatoid?
By the same token, paragraph 2.2. “Patients with mesothelioma”: what was the histological subtype of the tumors from the pts providing pleural effusion or tumor biopsy?
Line 99, “based on TCGA”: the abbreviation should be defined here, not on line 100 as it is the first time it is used, not on line 100.
Line 114-116, “GEPIA (http://gepia.cancer-pku.cn/) is a web server for cancer and normal gene expression profiling and interactive analysis”: Wouldn't be better to describe it on line 97, i.e. the 1st time GEPIA2 is mentioned?
Line 132: Fig. 3A is cited before figure 2.
Line 141-142, “We selected the well-known genes as representatives to analyze their correlation with EMT gene signature”: In which respect are the genes well-known? Maybe the authors should have simply written "well-known genes" (w/o “the”) such as those listed underneath? Please clarify/rephrase.
RESULTS
Line 158-160, “MESO EMT gene signature was generated from the overlaps of up-regulated genes at all time points of tumor challenge by microarray and scRNA-Seq analysis in mouse models”: The sentence needs clarifications. The authors ought to specify here in 3.1. what are the time points (in Mat. & Meth. it is mentioned that 5 mice were sacrificed weekly over an 8-week time period. Are the time points so every week? Or are those indicated in 3.2. line 202-203?). Moreover, it is stated in paragraph 2.2. of Math. & Meth. that scRNA-seq analysis was carried on pleural effusions, which were derived from institutional patients, not mouse models, as written on line 158-160.
Line 178-179, “Notably, the more powerful genes SERPINH1, SPARC …”: perhaps better to describe them as most prognostic genes, as the term “powerful” can be misinterpreted.
Line 180-183, “The AUC is greater than 0.98 when combining SERPINH1 and SPARC, however, the prognostic value did not increase dramatically when AUC reaches 0.99 at these 4 genes combination, supporting an idea that the top genes could perform excellently on their own or in combination”: the statement raises the question of what is the AUC for each of the 9 genes? If they each perform excellently on their own, they should show similar AUC.
FIGURE 1
Figure1A is described as representing data obtained in mouse models. Are these models derived from RN5 cells transplanted in mice (the only mouse model described in Mat. & Meth.)? It should be specified both in paragraph 3.1. of Results and in the figure legend.
And are figure 1B-D derived from mice or pts? N in each figure is 41 for low and high groups (if discerned well, as it is barely readable). Please explain where these cases come from. TCGA? Own institution? The same applies to figures 1S and 2S
Figure 1C: the text in each graph is barely legible in the pdf file provided for review, hopefully the original figures have better resolution and are more suitable for publication. Furthermore, all the KM curves diverge a bit late, as if the prognostic value for each EMT gene is not manifest initially. For ex., the prognostic effect of CALD1 and TIMP3 (and perhaps also ITGV6) is not visible before 40 months, when <30% of pts is alive, thus questioning the clinical utility of these genes. What would be the prognostic value of each gene over a period of 24 months, which more realistically reflects the median OS of MPM pts?
Line 194-195, figure legend B) and C), “Survival curve of 9 gene signature” and “Survival curves of each EMT gene”, respectively: it sounds like it is the 9 gene signature or each EMT gene that is surviving. The sentence should be reformulated more properly (for ex., "survival curves based on high/low expression of the 9 gene signature” or “… of each EMT gene" or something similar).
Line 199, “The AUC of ROC curve is 1”: presumably this refers to the entire 9-gene signature, and if so, it should be specified.
Line 207-208, “tissues from naïve patients with MESO”: naïve for what, therapy-naïve?
DATA in FIGURE 2
Legend to figure 2, line 219-220, “C&D) scRNA-Seq results acquired from MESO patients with pleural effusion (n=3), and E&F) Biopsy tumor tissues from patients with MESO (n=5)”: it should be specified here (and/or in Mat. & Meth.) whether effusions and biopsies were obtained from the same MESO pts or different pts.
Line 208-209, “these EMT genes were overexpressed predominantly on mesothelioma cells characterized by MSLN expression”. Figure 2C-D: the cell types corresponding to 0-13 should be specified in the figures. It should also be indicated, which are the MSLN-expressing cells, presumably cell population 6. Do the authors mean that only MSLN-positive tumor cells expressed EMT genes or that the cells expressing EMT genes where MESO cells, which can be identified by MSLN expression? MSLN is known to be particularly expressed in epithelioid MESO, not so much in sarcomatoid MESO, thus the need for sentence clarification and specification of the histological MESO subtype analyzed in the figure.
By the same token, which are the MSLN-expressing tumor cells in fig 2E-F? Cancer cells? Please also indicate the presumed CAF population in 2E-F.
Line 209-213, “Another cluster could be detected in the tumor tissues, but not in the pleural effusion, which may represent cancer-associated fibroblast (CAF) population with low expression of MSLN and WT1 but with high expression of ACTA2, S100A4, COL5A2, SPARC and FN1”: the authors say that this cluster may (thus not certain) represent CAFs, based on gene expression. Have the authors validated by histology and IHC whether this CAF population exist in the MESO samples that they have analyzed?
FIGURE 3
What is N for high and low scores in figures 3B, C and D? Are the KM curves generated using the TCGA data set?
Line 255-256, “GO enrichment analysis for the EMT genes in BP, CC and MF terms”: for clarity the abbreviations should be defined in the text too, as they are in fig. 4A.
Entire paragraph 3.4., line 252-269: The authors deserve definitely credit for the nice correlation of EMT vs activation of carcinogenic pathways and suppression of immune response via down-regulation of IFN-alpha and IFN-gamma pathway, upregulation of TGFB1/TGFBR1 as well as activation of ICP proteins!!
Paragraph 3.5., “Correlation of MESO EMT genes with genomic/epigenomic alterations”: again, the results should be better explained. Where were these correlations obtained from? On the pts tissue samples or from TCGA data? The same questions apply to figures 6S, 7S and 8S: are the data shown derived from the TCGA db?
FIGURE 5
The figure seems to show correlations only with gene expression. Have the authors tried to correlate EMT genes with mutations, fusions and splice alterations of the BAP1, NF2 and SETD2 genes, which occur in a substantial proportion of MESO (Figure 8S of the MS, Bueno et al. 2016 in reference xlii, and Hmeljak j et al. 2018 Cancer Discov. 2018 doi: 10.1158/2159-8290.CD-18-0804)?.
DISCUSSION
Line 332-334, “It was also demonstrated that EMT was induced by various signaling pathways, including TGFB1, Wnt-B-catenin, NOTCH and receptor tyrosine kinases”: This is not really shown in this study. Either the authors show these data or cite the work where this was shown or remove the sentence.
TGFB1 has both tumor-suppressive and -promoting effects, also depending on the cell type. This seems to be the case for MESO too. For ex., in Li B et al. BMC Cancer 2012 doi: 10.1186/1471-2407-12-239, TGFB1 is shown to have suppressive effect on MESO cell lines. It has been hypothesized that TGFB1 might have a different effect on epithelioid and sarcomatoid MESO cells. For completeness, the authors should comment on these bivalent aspects of TGFB1.
Line 349-350, “Distinct feature of EMT gene expression included enrichment of TILs and overexpression of immunoinhibitors”: on line 266-267, describing fig. 4C, the authors state that "the expression of MESO-EMT genes is associated with down-regulation of activated CD8 TILs". This seems to be in contradiction with the enrichment of TILs mentioned on line 349-350 and should be clarified.
Line 357-359, “CAFs isolated from lung cancer promoted EMT via producing stromal cell-derived factor-1 (SDF-1) by upregulating CXCR4 and β-catenin”: it would have been better to write “thereby upregulating CXCR4 and beta-catenin”. Indeed, the citation XXXiX shows that CAFs produce SDF-1 and induce EMT of lung adenocarcinoma via CXCR4/β-catenin/PPARδ signalling.
Line 367-369, “Comprehensive genomic analysis of MESO demonstrated recurrent mutations, gene fusions and splicing alterations in MESO across different trialsxlii,xliii,xliv. Gene fusions and splice alterations are likely to be the underlying mechanisms resulting in gene inactivation of NF2, BAP1 and SETD2”: as also shown in figure 8S, mutations and deletions are the most common mechanisms of inactivation for these three genes.
The authors should discuss what are the new achievements of the current study as compared to their recent publication on EMT genes in MESO (reference xvi).
References, more as editorial aspect to be aware of: References are in Roman numerals, but when they are recited in the text, Arabic numerals are used instead. The reason for that is not explained, although it seems intuitive. In any case, it does not seem to conform with IJMS’s reference/citation style (Arabic numerals placed in square brackets).
Moreover, journal names are sometimes in italics sometimes not in italics. Does it mean anything particular or just that it needs to be adjusted and made consistent throughout the list of references?
If the authors could address most of the comments, I would be happy to evaluate a revised version of this promising and quite interesting MS. Thank you for the opportunity to review it.
Author Response
We would like to thank you and the reviewers for taking the time to review our manuscript and for the constructive comments.
We have revised according to the comments and answered the reviewers point-by-point.
Reviewer #1:
We would like to thank the reviewer for taking the time to review our work and for the constructive comments. We have modified the manuscript accordingly.
This is an interesting MS showing strong correlation between the expression of a 9 EMT gene signature and poor prognosis in malignant mesothelioma (abbreviated as MESO by the Authors). The study in large part is a follow-up of a recent publication by the same research group (reference xvi). The MESO EMT gene signature was defined by transcriptomics analysis of overlapping up-regulated genes (as assessed by GeneVenn diagrams) in a mouse model (evaluated by microarrays) and in pleural effusions of MESO patients from the authors’ institution (evaluated by scRNA-seq). Utilizing multiomics analyses and several databases, the Authors display the prognostic value and correlation with OS of the MESO EMT gene signature, which cells express the 9 EMT genes (allegedly MESO cells and CAFs), and that EMT gene expression is higher in non-epithelioid than epithelioid MESO samples of the TCGA db. Moreover, to explain the negative prognostic impact of the 9 EMT gene signature, they show that it correlates with the hypermethylation of certain epigenetic genes and with the loss of CDKN2A/B expression, and that it is linked to important cancer-relevant pathways and an immunosuppressive microenvironment.
However, some of the methods and data need clarifications and adjustments before the MS is fully suitable for publication.
Thank you very much for your comments.
SPECIFIC POINTS
MATERIALS AND METHODS
As remarked by the Authors, MESO is a cancer type with strong tendency to EMT, as proven by its 3 main histological subtypes, including the sarcomatoid with fully occurred EMT. Thus, the study carried out by Wu et al. is certainly relevant both for the understanding of EMT pathogenesis in cancer in general and in MESO, and for identifying possible vulnerabilities in this process that may represent novel potential therapeutic targets. In this respect, a more detailed histopathological description of the utilized samples is missing. It is only mentioned that the murine MESO cell line RN5 had biphasic morphology. Did the tumors that grew in mice after i.p. injection maintain a biphasic histology or did they become sarcomatoid?
-In the peritoneal model, we collected the single cells in the peritoneal cavity after tumor cell ip injection, and looked at gene profiles of all cells in tumor microenvironment. In our previous experiments, we have seen that RN5 tumors remain characterized by a biphasic phenotype, and did not become sarcomatoid.
By the same token, paragraph 2.2. “Patients with mesothelioma”: what was the histological subtype of the tumors from the pts providing pleural effusion or tumor biopsy?
-The histological subtype of the MESO patients with PE (n=3 epithelioid). Whereas the biopsy tumors (n=5) were diagnosed as 1 biphasic and 4 epithelioid subtypes.
Line 99, “based on TCGA”: the abbreviation should be defined here, not on line 100 as it is the first time it is used, not on line 100.
-The full name of TCGA has been spelled out on line 99: TCGA (The Cancer Genome Atlas).
Line 114-116, “GEPIA (http://gepia.cancer-pku.cn/) is a web server for cancer and normal gene expression profiling and interactive analysis”: Wouldn't be better to describe it on line 97, i.e. the 1st time GEPIA2 is mentioned?
- We moved “GEPIA is a web server for cancer and normal gene expression profiling and interactive analysis.” to line 97 as suggested.
Line 132: Fig. 3A is cited before figure 2.
- We deleted Fig. 3A in line 132 and cited Ref #19 for EMT score calculation.
Line 141-142, “We selected the well-known genes as representatives to analyze their correlation with EMT gene signature”: In which respect are the genes well-known? Maybe the authors should have simply written "well-known genes" (w/o “the”) such as those listed underneath? Please clarify/rephrase.
-We totally agree with you to remove “the”, changing into “well-known genes”.
RESULTS
Line 158-160, “MESO EMT gene signature was generated from the overlaps of up-regulated genes at all time points of tumor challenge by microarray and scRNA-Seq analysis in mouse models”: The sentence needs clarifications. The authors ought to specify here in 3.1. what are the time points (in Mat. & Meth. it is mentioned that 5 mice were sacrificed weekly over an 8-week time period. Are the time points so every week? Or are those indicated in 3.2. line 202-203?).
-We added a few sentences to describe the methods in Mat. & Meth. Murine mesothelioma RN5 cells were injected ip. into mice, and 5 mice were sacrificed weekly over an 8-week time period. Time points included 1, 2, 3, 4, 5 and 8 weeks.
We also made some change in Fig. 1 legend: Venn diagram schema illustrates how to identify the EMT gene signature in mouse mesothelioma, where a-f represents 6 time points compared with naïve mice, and g represents the gene list generated from scRNA-Seq data (4weeks vs naïve).
Moreover, it is stated in paragraph 2.2. of Math. & Meth. that scRNA-seq analysis was carried on pleural effusions, which were derived from institutional patients, not mouse models, as written on line 158-160.
-Yes, clinical samples were collected from MESO patient with pleural effusion and therapy-naïve patients. We performed scRNA-Seq analysis to demonstrate the EMT genes in patient samples.
Line 178-179, “Notably, the more powerful genes SERPINH1, SPARC …”: perhaps better to describe them as most prognostic genes, as the term “powerful” can be misinterpreted.
-Thanks for your suggestion. We changed “the more powerful genes” into “the most prognostic genes” as you recommended.
Line 180-183, “The AUC is greater than 0.98 when combining SERPINH1 and SPARC, however, the prognostic value did not increase dramatically when AUC reaches 0.99 at these 4 genes combination, supporting an idea that the top genes could perform excellently on their own or in combination”: the statement raises the question of what is the AUC for each of the 9 genes? If they each perform excellently on their own, they should show similar AUC.
-This is really a good question. We analyzed AUC from different ways, either from the most prognostic gene SERPINH1 or least prognostic gene TIMP3. Clear changes could be observed. As mentioned, the AUC is greater than 0.98 when combining SERPINH1 and SPARC, supporting an idea that the top genes could perform excellently on their own or in combination. Each individual gene from the signature may have a strong prognostic value. Supplementary data Fig. 1S & 2S may help overview the prognostic value of each gene.
FIGURE 1
Figure1A is described as representing data obtained in mouse models. Are these models derived from RN5 cells transplanted in mice (the only mouse model described in Mat. & Meth.)? It should be specified both in paragraph 3.1. of Results and in the figure legend.
And are figure 1B-D derived from mice or pts? N in each figure is 41 for low and high groups (if discerned well, as it is barely readable). Please explain where these cases come from. TCGA? Own institution? The same applies to figures 1S and 2S
-We used the RN5 mouse model in this study. This point was clarified in the Mat. & Meth and in the Results, and in the figure legend.
We added a few sentences to describe the methods in Mat. & Meth. Murine mesothelioma RN5 cells were injected ip. into mice, and 5 mice were sacrificed weekly over an 8-week time period. Time points included 1, 2, 3, 4, 5 and 8 weeks.
We also made some change in Fig. 1 legend: Venn diagram schema illustrates how to identify the EMT gene signature in mouse mesothelioma, where a-f represents 6 time points compared with naïve mice, and g represents the gene list generated from scRNA-Seq data (4weeks vs naïve).
-Fig. 1B-D were generated from the TCGA database, similarly to Fig. 1S & 2S.
Figure 1C: the text in each graph is barely legible in the pdf file provided for review, hopefully the original figures have better resolution and are more suitable for publication. Furthermore, all the KM curves diverge a bit late, as if the prognostic value for each EMT gene is not manifest initially. For ex., the prognostic effect of CALD1 and TIMP3 (and perhaps also ITGV6) is not visible before 40 months, when <30% of pts is alive, thus questioning the clinical utility of these genes. What would be the prognostic value of each gene over a period of 24 months, which more realistically reflects the median OS of MPM pts?
-Each gene has significant prognostic value in the MESO cohort of TCGA database, as shown in Fig. 1C. However, the least significant genes such as CALD1 or TIMP3 had a delayed impact on survival and the separation of the survival curves was seen only after 30-40 months. On the other hand, the most significant genes such as SERPINH1, SPARC, COL5A2, TNFRSF12A and ACTA2 have an immediate impact on survival with early separation of the survival curves and are thus more relevant.
Line 194-195, figure legend B) and C), “Survival curve of 9 gene signature” and “Survival curves of each EMT gene”, respectively: it sounds like it is the 9 gene signature or each EMT gene that is surviving. The sentence should be reformulated more properly (for ex., "survival curves based on high/low expression of the 9 gene signature” or “… of each EMT gene" or something similar).
-We modified the figure legends in Fig. 1B & 1C, “Survival curve based on expression level of 9 gene signature, …
Line 199, “The AUC of ROC curve is 1”: presumably this refers to the entire 9-gene signature, and if so, it should be specified.
- The AUC of ROC curve is 1, representing showing that there is a cutoff for which both sensitivity and specificity are at 100%, not referring to entire 9 gene signature, which the AUC of ROC cure is very close to 1 as shown in Fig. 1S & 2S.
Line 207-208, “tissues from naïve patients with MESO”: naïve for what, therapy-naïve?
-We have changed “tissues from naïve patients with MESO” into “tissues from therapy-naïve patients with MESO”.
DATA in FIGURE 2
Legend to figure 2, line 219-220, “C&D) scRNA-Seq results acquired from MESO patients with pleural effusion (n=3), and E&F) Biopsy tumor tissues from patients with MESO (n=5)”: it should be specified here (and/or in Mat. & Meth.) whether effusions and biopsies were obtained from the same MESO pts or different pts.
-We added more information and explained where the single cells were obtained. The pleural effusion was obtained from the same patients as the biopsy when possible.
The manuscript was clarified: “Samples were acquired from tumor biopsy (n=5) and from the pleural effusion (n=3). PE and tumor biopsy samples came from different patients.” Were added in Mat. & Meth. line 85-87.
Line 208-209, “these EMT genes were overexpressed predominantly on mesothelioma cells characterized by MSLN expression”. Figure 2C-D: the cell types corresponding to 0-13 should be specified in the figures. It should also be indicated, which are the MSLN-expressing cells, presumably cell population 6. Do the authors mean that only MSLN-positive tumor cells expressed EMT genes or that the cells expressing EMT genes where MESO cells, which can be identified by MSLN expression? MSLN is known to be particularly expressed in epithelioid MESO, not so much in sarcomatoid MESO, thus the need for sentence clarification and specification of the histological MESO subtype analyzed in the figure.
By the same token, which are the MSLN-expressing tumor cells in fig 2E-F? Cancer cells? Please also indicate the presumed CAF population in 2E-F.
- We more specifically focused on tumor cells that were characterized by MSLN expression, but did not exclude other clusters that may express EMT genes. Fig. 2D, 2E, 2F are presented to show the EMT gene expression in each cluster.
Line 209-213, “Another cluster could be detected in the tumor tissues, but not in the pleural effusion, which may represent cancer-associated fibroblast (CAF) population with low expression of MSLN and WT1 but with high expression of ACTA2, S100A4, COL5A2, SPARC and FN1”: the authors say that this cluster may (thus not certain) represent CAFs, based on gene expression. Have the authors validated by histology and IHC whether this CAF population exist in the MESO samples that they have analyzed?
-This is a great question. From CAF markers ACTA2, S100A4, COL5A2, SPARC and FN1, we could identify the other cluster of cancer cells that express low MSLN and WT1 but high CAF marker, suggesting that they might be CAF population. Unfortunately, we have not demonstrated by IHC at present. We will investigate the CAF population in MESO in the future, as this is an important population in tumor microenvironment.
FIGURE 3
What is N for high and low scores in figures 3B, C and D? Are the KM curves generated using the TCGA data set?
-Fig. 3B and KM curves were both generated from TCGA data set, EMT score Hi n=44, EMT score Lo n=43. TCGA data set (n=87, including 62 epithelioid and 25 non-epitheloid subtypes) were divided into high and low groups based on EMT score of each patient.
We added a sentence in Methods on line 123-124: “EMT score results were generated from TCGA data set”.
Line 255-256, “GO enrichment analysis for the EMT genes in BP, CC and MF terms”: for clarity the abbreviations should be defined in the text too, as they are in fig. 4A.
- GO enrichment analysis for the EMT genes in BP, CC and MF terms were spelled out in full text in Fig. 4A and 3.4 line 263-264 Gene ontology (GO) enrichment analysis for the EMT genes in biological process (BP), cellular component (CC) and molecular function (MF).
Entire paragraph 3.4., line 252-269: The authors deserve definitely credit for the nice correlation of EMT vs activation of carcinogenic pathways and suppression of immune response via down-regulation of IFN-alpha and IFN-gamma pathway, upregulation of TGFB1/TGFBR1 as well as activation of ICP proteins!!
-Thank you for the comments. The results shown in Fig. 4 were based on omics analysis platform. Overall, EMT gene expression is positively correlated with immunosuppressive pathways but negatively correlated with immunostimulatory pathways. On protein level, proteomic analysis was performed to look at protein-protein interaction (PPI) of EMT genes and network they may be involved, please see Fig. 5S for reference.
Paragraph 3.5., “Correlation of MESO EMT genes with genomic/epigenomic alterations”: again, the results should be better explained. Where were these correlations obtained from? On the pts tissue samples or from TCGA data? The same questions apply to figures 6S, 7S and 8S: are the data shown derived from the TCGA db?
-Gene expression data of EMT, genomic and epigenomic alterations were retrieved from MESO TCGA cbioportal, as described in methods. We then analyzed the correlation between each gene set, and the heatmap was plotted using prism software, the same as Fig. 7S & 8S. In Fig. 6S, the correlation between EMT gene expression and immunoinhibitor gene in MESO was obtained from multiple heterogeneous data sets (http://cis.hku.hk/TISIDB/).
FIGURE 5
The figure seems to show correlations only with gene expression. Have the authors tried to correlate EMT genes with mutations, fusions and splice alterations of the BAP1, NF2 and SETD2 genes, which occur in a substantial proportion of MESO (Figure 8S of the MS, Bueno et al. 2016 in reference xlii, and Hmeljak j et al. 2018 Cancer Discov. 2018 doi: 10.1158/2159-8290.CD-18-0804)?.
-This figure only presented correlations with gene expression. As shown in supplementary data, EMT genes has very rare mutations (Fig. 7S), whereas gene mutations are more common in BAP1, NF2, STED2 and CDKN2A/B (Fig. 8SA-D). Perhaps we could simply compare the rate of mutations. The difference is quite clear.
DISCUSSION
Line 332-334, “It was also demonstrated that EMT was induced by various signaling pathways, including TGFB1, Wnt-B-catenin, NOTCH and receptor tyrosine kinases”: This is not really shown in this study. Either the authors show these data or cite the work where this was shown or remove the sentence.
-Due to space limitation, Fig. 4 only selected top 10 pathways with up- or down-regulation. We have removed Wnt B catenin, NOTCH and receptor tyrosine kinase to avoid any confusion as suggested.
TGFB1 has both tumor-suppressive and -promoting effects, also depending on the cell type. This seems to be the case for MESO too. For ex., in Li B et al. BMC Cancer 2012 doi: 10.1186/1471-2407-12-239, TGFB1 is shown to have suppressive effect on MESO cell lines. It has been hypothesized that TGFB1 might have a different effect on epithelioid and sarcomatoid MESO cells. For completeness, the authors should comment on these bivalent aspects of TGFB1.
-We added a sentence on this point on p12 line 346-347. “The role of TGF-β as a primary inducer of EMT has been reported in several cancers, while other studies reported a more ambivalent role for TGF-b1.”
Line 349-350, “Distinct feature of EMT gene expression included enrichment of TILs and overexpression of immunoinhibitors”: on line 266-267, describing fig. 4C, the authors state that "the expression of MESO-EMT genes is associated with down-regulation of activated CD8 TILs". This seems to be in contradiction with the enrichment of TILs mentioned on line 349-350 and should be clarified.
-On line 349-350: we clarified this sentence, “Distinct feature of EMT gene expression included down-regulation of CD8 TILs and overexpression of immunoinhibitors.”
Line 357-359, “CAFs isolated from lung cancer promoted EMT via producing stromal cell-derived factor-1 (SDF-1) by upregulating CXCR4 and β-catenin”: it would have been better to write “thereby upregulating CXCR4 and beta-catenin”. Indeed, the citation XXXiX shows that CAFs produce SDF-1 and induce EMT of lung adenocarcinoma via CXCR4/β-catenin/PPARδ signalling.
-We modified as “CAFs isolated from lung cancer promoted EMT via producing stromal cell-derived factor-1 (SDF-1) thereby upregulating CXCR4 and β-catenin,…”
Line 367-369, “Comprehensive genomic analysis of MESO demonstrated recurrent mutations, gene fusions and splicing alterations in MESO across different trialsxlii,xliii,xliv. Gene fusions and splice alterations are likely to be the underlying mechanisms resulting in gene inactivation of NF2, BAP1 and SETD2”: as also shown in figure 8S, mutations and deletions are the most common mechanisms of inactivation for these three genes.
-Thanks for pointing this out, to make this statement more accurate, we changed into: “Gene fusions and deep deletions are likely to be the underlying mechanisms resulting in gene inactivation of NF2, BAP1 and SETD2”.
The authors should discuss what are the new achievements of the current study as compared to their recent publication on EMT genes in MESO (reference xvi).
-On p13 line 392-393, we added a few more discussions here, “Taken altogether, better understanding of EMT gene profiling provided by this study will help to establish the linkage between EMT and epigenetic mechanisms.”
References, more as editorial aspect to be aware of: References are in Roman numerals, but when they are recited in the text, Arabic numerals are used instead. The reason for that is not explained, although it seems intuitive. In any case, it does not seem to conform with IJMS’s reference/citation style (Arabic numerals placed in square brackets).
Moreover, journal names are sometimes in italics sometimes not in italics. Does it mean anything particular or just that it needs to be adjusted and made consistent throughout the list of references?
-Reference information was copied from citation format. References editing could be conducted depending on specific requirements of journal editors.
If the authors could address most of the comments, I would be happy to evaluate a revised version of this promising and quite interesting MS. Thank you for the opportunity to review it.

Reviewer 2 Report
The manuscript is difficult to follow, many aspects are given by implication and not adequately explained. The starting data, already described in a previous paper, are not presented clearly. For example: what are the time points described in figure 1? what do the letters within the venn diagrams in figure 1A mean? Why did the authors choose to consider only upregulated genes?
The method of defining and evaluating the prognostic signature is not precisely described and a ROC curve with 100% specificity and 100% sensitivity is hard to believe. How was it achieved? How many months of follow-up were considered for ROC curve analysis?
The usefulness of the different final scores remains unclear.
A GO analysis performed on EMT-gene is certainly biased.
The conclusions are hasty and not adequately supported by the data presented.
Author Response
We would like to thank you and the reviewers for taking the time to review our manuscript and for the constructive comments.
We have revised according to the comments and answered the reviewers point-by-point.
Reviewer #2:
We would like to thank the reviewer for taking the time to review our work and for the constructive comments. We have modified the manuscript accordingly.
The manuscript is difficult to follow, many aspects are given by implication and not adequately explained. The starting data, already described in a previous paper, are not presented clearly. For example: what are the time points described in figure 1? what do the letters within the venn diagrams in figure 1A mean?
-On page 4: We clarified as “MESO EMT gene signature was generated from the overlaps of up-regulated genes at all time points of tumor challenge by microarray at 1, 2, 3, 4, 5, 8 weeks timepoints (a-f) and scRNA-Seq analysis at 4 weeks (g) after tumor cell ip injection in mouse models.”
Changes were made in Methods and figure legend as well.
Why did the authors choose to consider only upregulated genes?
-This is because the up-regulated genes are preferentially associated with tumor growth and immunosuppression. We already know that EMT process is able to enhance tumor invasion and metastasis, cell cycle, stemness, and so on. Therefore, the up-regulated genes are more likely the candidates involved in the EMT process. Also upregulated genes may provide an opportunity for choosing new therapeutic targets to block the EMT process.
The method of defining and evaluating the prognostic signature is not precisely described and a ROC curve with 100% specificity and 100% sensitivity is hard to believe. How was it achieved? How many months of follow-up were considered for ROC curve analysis?
The usefulness of the different final scores remains unclear.
A GO analysis performed on EMT-gene is certainly biased.
The conclusions are hasty and not adequately supported by the data presented.
-These clinical data were retrieved from TCGA database (https://www.cbioportal.org/datasets).
As stated in the methods, the ROC curve was generated by using the most significant gene SERPINH1 and then adding one gene at a time to calculate the areas under ROC curves up to 9 genes. The AUC of ROC curve is 1 when cut-off for which both sensitivity and specificity are at 100%. This demonstrates that the use of these 9 genes can reliably define the EMT process. The survival data showed that the surviving time was as long as 84 months.
-The gene signature reported here was identified in our animal model and some of our patients. The value of the gene signature was then demonstrated in the TCGA database (87 cases: 62 epithelioid vs 25 non-epithelioid subtypes). The importance of these genes will need to be confirmed in further studies, but for the first we obtained a gene signature that specifically and reliably identified EMT in mesothelioma. We therefore believe that these results are meaningful and valuable for MESO both in terms of prognosis and to identify potential therapeutic targets to block the EMT process.

Round 2
Reviewer 1 Report
Review of the revised MS ijms-1922496
The authors have successfully addressed several but not most/all of my initial comments. There still remain the following points (most are minor) to adjust and/or clarify (indicated with initial comment, reply from the Authors in italics and my new comment in bold):
paragraph 2.2. “Patients with mesothelioma”: what was the histological subtype of the tumors from the pts providing pleural effusion or tumor biopsy?
-The histological subtype of the MESO patients with PE (n=3 epithelioid). Whereas the biopsy tumors (n=5) were diagnosed as 1 biphasic and 4 epithelioid subtypes.
I’d suggest that the Authors specify this in paragraph 2.2, i.e., PE (n=3, all with epithelioid MESO cells); biopsy tumor tissues (n=5; 1 biphasic and 4 epithelioid subtypes). As note of caution, in PE basically only epithelioid MESO cells can be found, as sarcomatoid MESO cells are sitting deep in the pleura and do not usually exfoliate. Thus, as indicated above, I suggest using the term PE with epithelioid MESO cells (i.e., they could be coming from epithelioid or biphasic MESO), rather than categorically classifying the MESO identified by examination of PE as epithelioid MESO.
Line 141-142, “We selected the well-known genes as representatives to analyze their correlation with EMT gene signature”: In which respect are the genes well-known? Maybe the authors should have simply written "well-known genes" (w/o “the”) such as those listed underneath? Please clarify/rephrase.
-We totally agree with you to remove “the”, changing into “well-known genes”.
I don’t see that this has been done, the sentence (now on line 146-147 of the pdf file) is still written as “We selected the well-known genes as representatives …”.
RESULTS
Line 158-160, “MESO EMT gene signature was generated from the overlaps of up-regulated genes at all time points of tumor challenge by microarray and scRNA-Seq analysis in mouse models”: The sentence needs clarifications. The authors ought to specify here in 3.1. what are the time points (in Mat. & Meth. it is mentioned that 5 mice were sacrificed weekly over an 8-week time period. Are the time points so every week? Or are those indicated in 3.2. line 202-203?).
-We added a few sentences to describe the methods in Mat. & Meth. Murine mesothelioma RN5 cells were injected ip. into mice, and 5 mice were sacrificed weekly over an 8-week time period. Time points included 1, 2, 3, 4, 5 and 8 weeks.
It’s a bit contradictory, because if it’s weekly over 8 weeks, there should be 8 time-points, not 6. The authors could simplify by writing that 5 mice were sacrificed at 6 time-points over an 8-week time-period (week 1, 2, 3, 4, 5 and 8).
We also made some change in Fig. 1 legend: Venn diagram schema illustrates how to identify the EMT gene signature in mouse mesothelioma, where a-f represents 6 time points compared with naïve mice, and g represents the gene list generated from scRNA-Seq data (4weeks vs naïve).
It is no clear what the “naïve mice” are in this context, especially because they are used as comparators (“where a-f represents 6 time points compared with naïve mice”). The “naïve mice” should be explained or described with different terms.
Moreover, it is stated in paragraph 2.2. of Math. & Meth. that scRNA-seq analysis was carried on pleural effusions, which were derived from institutional patients, not mouse models, as written on line 158-160.
-Yes, clinical samples were collected from MESO patient with pleural effusion and therapy-naïve patients. We performed scRNA-Seq analysis to demonstrate the EMT genes in patient samples.
My original comment referred to the fact that in paragraph 3.1. and legend to Fig. 1 the Authors write that the “MESO EMT gene signature was generated from the overlaps of up-regulated genes at all time points of tumor challenge by microarray and scRNA-Seq analysis in mouse models”, while in paragraph 2.2. of Math. & Meth. It is stated that scRNA-seq analysis was carried on PEs from patients. Therefore, it should be clarified whether scRNA-seq analysis was performed only on the 3 PEs from patients or also on the samples from the mouse model.
FIGURE 1
Figure1A is described as representing data obtained in mouse models. Are these models derived from RN5 cells transplanted in mice (the only mouse model described in Mat. & Meth.)? It should be specified both in paragraph 3.1. of Results and in the figure legend. And are figure 1B-D derived from mice or pts? N in each figure is 41 for low and high groups (if discerned well, as it is barely readable). Please explain where these cases come from. TCGA? Own institution? The same applies to figures 1S and 2S
-We used the RN5 mouse model in this study. This point was clarified in the Mat. & Meth and in the Results, and in the figure legend.
We added a few sentences to describe the methods in Mat. & Meth. Murine mesothelioma RN5 cells were injected ip. into mice, and 5 mice were sacrificed weekly over an 8-week time period. Time points included 1, 2, 3, 4, 5 and 8 weeks.
We also made some change in Fig. 1 legend: Venn diagram schema illustrates how to identify the EMT gene signature in mouse mesothelioma, where a-f represents 6 time points compared with naïve mice, and g represents the gene list generated from scRNA-Seq data (4weeks vs naïve).
All these changes are fine, but the Authors keep writing in Math and Meth, Results and legend to Fig 1 that data were obtained in mouse models (plural), when the mouse model in fact is only one derived from RN5 cells. This is the change/clarification I was getting at.
-Fig. 1B-D were generated from the TCGA database, similarly to Fig. 1S & 2S.
Yes, but for clarity this should be indicated in the corresponding figure legends.
Figure 1C: the text in each graph is barely legible in the pdf file provided for review, hopefully the original figures have better resolution and are more suitable for publication. Furthermore, all the KM curves diverge a bit late, as if the prognostic value for each EMT gene is not manifest initially. For ex., the prognostic effect of CALD1 and TIMP3 (and perhaps also ITGV6) is not visible before 40 months, when <30% of pts is alive, thus questioning the clinical utility of these genes. What would be the prognostic value of each gene over a period of 24 months, which more realistically reflects the median OS of MPM pts?
-Each gene has significant prognostic value in the MESO cohort of TCGA database, as shown in Fig. 1C. However, the least significant genes such as CALD1 or TIMP3 had a delayed impact on survival and the separation of the survival curves was seen only after 30-40 months. On the other hand, the most significant genes such as SERPINH1, SPARC, COL5A2, TNFRSF12A and ACTA2 have an immediate impact on survival with early separation of the survival curves and are thus more relevant.
The reply to the comment is acceptable, however, to be cautious in their conclusions on the prognostic values of the genes in the EMT signature, the Authors should insert these statements in the Results (when they present Fig. 1C) and as a comment in the Discussion. As also demonstrated by the presented data in this study, most patients are dead after 40 months, thus it is difficult to envisage a clinical utility for the prognostic value of CALD1 and TIMP3.
Line 199, “The AUC of ROC curve is 1”: presumably this refers to the entire 9-gene signature, and if so, it should be specified.
- The AUC of ROC curve is 1, representing showing that there is a cutoff for which both sensitivity and specificity are at 100%, not referring to entire 9 gene signature, which the AUC of ROC cure is very close to 1 as shown in Fig. 1S & 2S.
It is still unclear how the AUC of the ROC curve = 1 was generated, as also mentioned by the other reviewer.
DATA in FIGURE 2
Legend to figure 2, line 219-220, “C&D) scRNA-Seq results acquired from MESO patients with pleural effusion (n=3), and E&F) Biopsy tumor tissues from patients with MESO (n=5)”: it should be specified here (and/or in Mat. & Meth.) whether effusions and biopsies were obtained from the same MESO pts or different pts.
-We added more information and explained where the single cells were obtained. The pleural effusion was obtained from the same patients as the biopsy when possible.
This does not seem to be true, as the Authors have added in Mat. & Meth. line 87-88 that “PE and tumor biopsy samples came from different patients”(as also specified in their reply underneath). Thus, the issue is still unclear.
The manuscript was clarified: “Samples were acquired from tumor biopsy (n=5) and from the pleural effusion (n=3). PE and tumor biopsy samples came from different patients.” Were added in Mat. & Meth. line 85-87.
In the revised MS available for review, the sentence “Samples were acquired from tumor biopsy (n=5) and from the pleural effusion (n=3)” appears deleted. Re the other sentence, see above.
Line 208-209, “these EMT genes were overexpressed predominantly on mesothelioma cells characterized by MSLN expression”. Figure 2C-D: the cell types corresponding to 0-13 should be specified in the figures.
- We more specifically focused on tumor cells that were characterized by MSLN expression, but did not exclude other clusters that may express EMT genes. Fig. 2D, 2E, 2F are presented to show the EMT gene expression in each cluster.
As suggested before, to be consistent with Fig 2E-F, the cell types corresponding to 0-13 in Fig 2C-D should be indicated in the two figures instead of numbers.
FIGURE 3
What is N for high and low scores in figures 3B, C and D? Are the KM curves generated using the TCGA data set?
-Fig. 3B and KM curves were both generated from TCGA data set, EMT score Hi n=44, EMT score Lo n=43. TCGA data set (n=87, including 62 epithelioid and 25 non-epitheloid subtypes) were divided into high and low groups based on EMT score of each patient.
Please indicate these values of N briefly in the legends to figure 3B-D.
TGFB1 has both tumor-suppressive and -promoting effects, also depending on the cell type. This seems to be the case for MESO too. For ex., in Li B et al. BMC Cancer 2012 doi: 10.1186/1471-2407-12-239, TGFB1 is shown to have suppressive effect on MESO cell lines. It has been hypothesized that TGFB1 might have a different effect on epithelioid and sarcomatoid MESO cells. For completeness, the authors should comment on these bivalent aspects of TGFB1.
-We added a sentence on this point on p12 line 346-347. “The role of TGF-β as a primary inducer of EMT has been reported in several cancers, while other studies reported a more ambivalent role for TGF-b1.”
On line 358-359, the Authors have also added the sentence “The bivalent aspects of TGFB1 have been addressed in some studies”: it appears incomplete and without the support of reference/references.
Line 367-369, “Comprehensive genomic analysis of MESO demonstrated recurrent mutations, gene fusions and splicing alterations in MESO across different trialsxlii,xliii,xliv. Gene fusions and splice alterations are likely to be the underlying mechanisms resulting in gene inactivation of NF2, BAP1 and SETD2”: as also shown in figure 8S, mutations and deletions are the most common mechanisms of inactivation for these three genes.
-Thanks for pointing this out, to make this statement more accurate, we changed into: “Gene fusions and deep deletions are likely to be the underlying mechanisms resulting in gene inactivation of NF2, BAP1 and SETD2”.
Still not completely correct. It should be “Mutations and deep deletions” not “Gene fusions and deep deletions”.
Author Response
Review of the revised MS ijms-1922496
The authors have successfully addressed several but not most/all of my initial comments. There still remain the following points (most are minor) to adjust and/or clarify (indicated with initial comment, reply from the Authors in italics and my new comment in bold):
paragraph 2.2. “Patients with mesothelioma”: what was the histological subtype of the tumors from the pts providing pleural effusion or tumor biopsy?
-The histological subtype of the MESO patients with PE (n=3 epithelioid). Whereas the biopsy tumors (n=5) were diagnosed as 1 biphasic and 4 epithelioid subtypes.
I’d suggest that the Authors specify this in paragraph 2.2, i.e., PE (n=3, all with epithelioid MESO cells); biopsy tumor tissues (n=5; 1 biphasic and 4 epithelioid subtypes). As note of caution, in PE basically only epithelioid MESO cells can be found, as sarcomatoid MESO cells are sitting deep in the pleura and do not usually exfoliate. Thus, as indicated above, I suggest using the term PE with epithelioid MESO cells (i.e., they could be coming from epithelioid or biphasic MESO), rather than categorically classifying the MESO identified by examination of PE as epithelioid MESO.
-Response 1: Paragraph 2.2 has been changed as you suggested, “Pleural effusions (PE, n=3, all with epithelioid MESO cells); and biopsy tumor tissues (n=5, 1 biphasic and 4 epithelioid subtypes). As a note of caution, in PE only epithelioid MESO cells can be found, as sarcomatoid MESO cells are sitting deep in the pleura and do not usually exfoliate. All samples were collected from MESO patients who were recruited between 2019 and 2020. PE and tumor biopsy samples came from the same patient (n=1) or from different patients (n=6).”
Line 141-142, “We selected the well-known genes as representatives to analyze their correlation with EMT gene signature”: In which respect are the genes well-known? Maybe the authors should have simply written "well-known genes" (w/o “the”) such as those listed underneath? Please clarify/rephrase.
-We totally agree with you to remove “the”, changing into “well-known genes”.
I don’t see that this has been done, the sentence (now on line 146-147 of the pdf file) is still written as “We selected the well-known genes as representatives …”.
-Response 2: We changed the sentence to: “We selected well-known genes to analyze their correlation with EMT gene signature. Selected genes included BAP1, NF2, TP53, LATS2, SETD2, CDKN2A, CDKN2B, PTCH1, KRAS and EGFR”.
RESULTS
Line 158-160, “MESO EMT gene signature was generated from the overlaps of up-regulated genes at all time points of tumor challenge by microarray and scRNA-Seq analysis in mouse models”: The sentence needs clarifications. The authors ought to specify here in 3.1. what are the time points (in Mat. & Meth. it is mentioned that 5 mice were sacrificed weekly over an 8-week time period. Are the time points so every week? Or are those indicated in 3.2. line 202-203?).
-We added a few sentences to describe the methods in Mat. & Meth. Murine mesothelioma RN5 cells were injected ip. into mice, and 5 mice were sacrificed weekly over an 8-week time period. Time points included 1, 2, 3, 4, 5 and 8 weeks.
It’s a bit contradictory, because if it’s weekly over 8 weeks, there should be 8 time-points, not 6. The authors could simplify by writing that 5 mice were sacrificed at 6 time-points over an 8-week time-period (week 1, 2, 3, 4, 5 and 8).
-Response 3: Thanks for your clarification. Line 81-83, “Five mice were sacrificed at 6 time-points over 8-week time-period (week 1, 2, 3, 4, 5 and 8). Five naive mice serving as controls.”
We also made some change in Fig. 1 legend: Venn diagram schema illustrates how to identify the EMT gene signature in mouse mesothelioma, where a-f represents 6 time points compared with naïve mice, and g represents the gene list generated from scRNA-Seq data (4weeks vs naïve).
It is no clear what the “naïve mice” are in this context, especially because they are used as comparators (“where a-f represents 6 time points compared with naïve mice”). The “naïve mice” should be explained or described with different terms.
-Response 4: Naïve mice are normal mice without tumor cell injection. This information was included in the legend from Fig 1.
Moreover, it is stated in paragraph 2.2. of Math. & Meth. that scRNA-seq analysis was carried on pleural effusions, which were derived from institutional patients, not mouse models, as written on line 158-160.
-Yes, clinical samples were collected from MESO patient with pleural effusion and therapy-naïve patients. We performed scRNA-Seq analysis to demonstrate the EMT genes in patient samples.
My original comment referred to the fact that in paragraph 3.1. and legend to Fig. 1 the Authors write that the “MESO EMT gene signature was generated from the overlaps of up-regulated genes at all time points of tumor challenge by microarray and scRNA-Seq analysis in mouse models”, while in paragraph 2.2. of Math. & Meth. It is stated that scRNA-seq analysis was carried on PEs from patients. Therefore, it should be clarified whether scRNA-seq analysis was performed only on the 3 PEs from patients or also on the samples from the mouse model.
-Response 5: scRNA seq was performed both in mouse samples and in patient samples.
In mouse intraperitoneal model, we performed scRNA-Seq analysis at the 4-week point.
For patients samples, we performed scRNA-Seq analysis using pleural effusion (n=3) and biopsy tumor tissues (n=5) from MESO patients.
We demonstrated that EMT genes identified in mouse model are also upregulated in patients with MESO from both pleural effusion and tumor tissue.
FIGURE 1
Figure1A is described as representing data obtained in mouse models. Are these models derived from RN5 cells transplanted in mice (the only mouse model described in Mat. & Meth.)? It should be specified both in paragraph 3.1. of Results and in the figure legend. And are figure 1B-D derived from mice or pts? N in each figure is 41 for low and high groups (if discerned well, as it is barely readable). Please explain where these cases come from. TCGA? Own institution? The same applies to figures 1S and 2S
-We used the RN5 mouse model in this study. This point was clarified in the Mat. & Meth and in the Results, and in the figure legend.
We added a few sentences to describe the methods in Mat. & Meth. Murine mesothelioma RN5 cells were injected ip. into mice, and 5 mice were sacrificed weekly over an 8-week time period. Time points included 1, 2, 3, 4, 5 and 8 weeks.
We also made some change in Fig. 1 legend: Venn diagram schema illustrates how to identify the EMT gene signature in mouse mesothelioma, where a-f represents 6 time points compared with naïve mice, and g represents the gene list generated from scRNA-Seq data (4weeks vs naïve).
All these changes are fine, but the Authors keep writing in Math and Meth, Results and legend to Fig 1 that data were obtained in mouse models (plural), when the mouse model in fact is only one derived from RN5 cells. This is the change/clarification I was getting at.
-Response 6: Fig. 1 data were obtained from mouse intraperitoneal (ip) model rather than intrapleural model. We clarified wherever it applies in the text. RN5 cells (2×106 /200 µl PBS) were injected intraperitoneally (ip) into the mice. The use of both mouse samples and patient samples for sc RNA seq was emphasized in the legend from Fig 1.
-Fig. 1B-D were generated from the TCGA database, similarly to Fig. 1S & 2S.
Yes, but for clarity this should be indicated in the corresponding figure legends.
-Response 7: We added “1B-D) were generated from the TCGA database.” before 1B in Fig. 1 legend, line 208.
Figure 1C: the text in each graph is barely legible in the pdf file provided for review, hopefully the original figures have better resolution and are more suitable for publication. Furthermore, all the KM curves diverge a bit late, as if the prognostic value for each EMT gene is not manifest initially. For ex., the prognostic effect of CALD1 and TIMP3 (and perhaps also ITGV6) is not visible before 40 months, when <30% of pts is alive, thus questioning the clinical utility of these genes. What would be the prognostic value of each gene over a period of 24 months, which more realistically reflects the median OS of MPM pts?
-Each gene has significant prognostic value in the MESO cohort of TCGA database, as shown in Fig. 1C. However, the least significant genes such as CALD1 or TIMP3 had a delayed impact on survival and the separation of the survival curves was seen only after 30-40 months. On the other hand, the most significant genes such as SERPINH1, SPARC, COL5A2, TNFRSF12A and ACTA2 have an immediate impact on survival with early separation of the survival curves and are thus more relevant.
The reply to the comment is acceptable, however, to be cautious in their conclusions on the prognostic values of the genes in the EMT signature, the Authors should insert these statements in the Results (when they present Fig. 1C) and as a comment in the Discussion. As also demonstrated by the presented data in this study, most patients are dead after 40 months, thus it is difficult to envisage a clinical utility for the prognostic value of CALD1 and TIMP3.
-Response 8: Lines 182-185: We inserted these statements in the Results when presenting Fig. 1C and as a comment in the Discussion, “As also demonstrated by the presented data in this study, most patients are dead after 40 months, thus it is difficult to envisage a clinical utility for the prognostic value of CALD1 and TIMP3.”
Line 199, “The AUC of ROC curve is 1”: presumably this refers to the entire 9-gene signature, and if so, it should be specified.
- The AUC of ROC curve is 1, representing showing that there is a cutoff for which both sensitivity and specificity are at 100%, not referring to entire 9 gene signature, which the AUC of ROC cure is very close to 1 as shown in Fig. 1S & 2S.
It is still unclear how the AUC of the ROC curve = 1 was generated, as also mentioned by the other reviewer.
-Response 9: To avoid confusion we replaced this sentence “The AUC of ROC curve is 1. This curve shows that there is a cutoff for which both sensitivity and specificity are at 100%.” by “The AUC of ROC curve is between 0.9-1, suggesting that both sensitivity and specificity are excellent.” in the supplementary data Fig. 1S &2S legends.
As known, the area under the ROC curve (AUC) results were considered excellent for AUC values between 0.9-1, good for AUC values between 0.8-0.9, fair for AUC values between 0.7-0.8, poor for AUC values between 0.6-0.7 and failed for AUC values between 0.5-0.6. Therefore, if the AUC of the ROC curve = 1, the curve shows that there is a cutoff for which both sensitivity and specificity are at 100%. Our results showed that AUC value of (TIMP3+CALD1) is 0.77878, which means fair, while AUC value of 9 genes is 0.9997, which means excellent.
DATA in FIGURE 2
Legend to figure 2, line 219-220, “C&D) scRNA-Seq results acquired from MESO patients with pleural effusion (n=3), and E&F) Biopsy tumor tissues from patients with MESO (n=5)”: it should be specified here (and/or in Mat. & Meth.) whether effusions and biopsies were obtained from the same MESO pts or different pts.
-We added more information and explained where the single cells were obtained. The pleural effusion was obtained from the same patients as the biopsy when possible.
This does not seem to be true, as the Authors have added in Mat. & Meth. line 87-88 that “PE and tumor biopsy samples came from different patients”(as also specified in their reply underneath). Thus, the issue is still unclear.
-Response 10: We clarified this point. Tumor and PE came from the same patient (n=1) or from different patients (n=6). Lines 87-89
The manuscript was clarified: “Samples were acquired from tumor biopsy (n=5) and from the pleural effusion (n=3). PE and tumor biopsy samples came from different patients.” Were added in Mat. & Meth. line 85-87.
In the revised MS available for review, the sentence “Samples were acquired from tumor biopsy (n=5) and from the pleural effusion (n=3)” appears deleted. Re the other sentence, see above.
-Response 11: AS noted above, we clarified this point. Tumor and PE came from the same patient (n=1) or from different patients (n=6). Lines 87-89
Line 208-209, “these EMT genes were overexpressed predominantly on mesothelioma cells characterized by MSLN expression”. Figure 2C-D: the cell types corresponding to 0-13 should be specified in the figures.
- We more specifically focused on tumor cells that were characterized by MSLN expression, but did not exclude other clusters that may express EMT genes. Fig. 2D, 2E, 2F are presented to show the EMT gene expression in each cluster.
As suggested before, to be consistent with Fig 2E-F, the cell types corresponding to 0-13 in Fig 2C-D should be indicated in the two figures instead of numbers.
-Response 12: We changed the cluster numbers of both PE and tumor biopsy data into cell types.
FIGURE 3
What is N for high and low scores in figures 3B, C and D? Are the KM curves generated using the TCGA data set?
-Fig. 3B and KM curves were both generated from TCGA data set, EMT score Hi n=44, EMT score Lo n=43. TCGA data set (n=87, including 62 epithelioid and 25 non-epitheloid subtypes) were divided into high and low groups based on EMT score of each patient.
Please indicate these values of N briefly in the legends to figure 3B-D.
-Response 13: The numbers of Fig 3B&D were included: (EMT score Hi n=43, EMT score Lo n=44); (Stromal score Hi n=43, Stromal score Lo n=44).
TGFB1 has both tumor-suppressive and -promoting effects, also depending on the cell type. This seems to be the case for MESO too. For ex., in Li B et al. BMC Cancer 2012 doi: 10.1186/1471-2407-12-239, TGFB1 is shown to have suppressive effect on MESO cell lines. It has been hypothesized that TGFB1 might have a different effect on epithelioid and sarcomatoid MESO cells. For completeness, the authors should comment on these bivalent aspects of TGFB1.
-We added a sentence on this point on p12 line 346-347. “The role of TGF-β as a primary inducer of EMT has been reported in several cancers, while other studies reported a more ambivalent role for TGF-b1.”
On line 358-359, the Authors have also added the sentence “The bivalent aspects of TGFB1 have been addressed in some studies”: it appears incomplete and without the support of reference/references.
-Response 14: Line 361 and 367, we added the references below:
- Morrison CD, Parvani JG, Schiemann WP. The relevance of the TGF-β Paradox to EMT-MET programs. Cancer Lett. 2013 Nov 28;341(1):30-40. doi: 10.1016/j.canlet.2013.02.048. Epub 2013 Mar 5. PMID: 23474494; PMCID: PMC3752409.
- Hao Y, Baker D, Ten Dijke P. TGF-β-Mediated Epithelial-Mesenchymal Transition and Cancer Metastasis. Int J Mol Sci. 2019 Jun 5;20(11):2767. doi: 10.3390/ijms20112767. PMID: 31195692; PMCID: PMC6600375.
- Shiota M, Fujimoto N, Matsumoto T, Tsukahara S, Nagakawa S, Ueda S, Ushijima M, Kashiwagi E, Takeuchi A, Inokuchi J, Uchiumi T, Eto M. Differential Impact of TGFB1Variation by Metastatic Status in Androgen-Deprivation Therapy for Prostate Cancer. Front Oncol. 2021 May 25;11:697955. doi: 10.3389/fonc.2021.697955. PMID: 34113577; PMCID: PMC8186782.
Line 367-369, “Comprehensive genomic analysis of MESO demonstrated recurrent mutations, gene fusions and splicing alterations in MESO across different trialsxlii,xliii,xliv. Gene fusions and splice alterations are likely to be the underlying mechanisms resulting in gene inactivation of NF2, BAP1 and SETD2”: as also shown in figure 8S, mutations and deletions are the most common mechanisms of inactivation for these three genes.
-Thanks for pointing this out, to make this statement more accurate, we changed into: “Gene fusions and deep deletions are likely to be the underlying mechanisms resulting in gene inactivation of NF2, BAP1 and SETD2”.
Still not completely correct. It should be “Mutations and deep deletions” not “Gene fusions and deep deletions”.
-Response 15: We corrected as you suggested, “Gene mutations and deep deletions…” line 391-392.
Submission Date
01 September 2022
Date of this review
02 Oct 2022 22:02:33

Reviewer 2 Report
-
Author Response
Thank you very much.